# Central nervous system hemangioblastomas in von Hippel-Lindau disease: Total growth rate and risk of developing new lesions not associated with circulating VEGF levels

Jimmy Sundblom[1]*, Tor Persson Skare[2], Olivia Holm[1], Staffan Welin[3],
Madelene Braun[4], Pelle Nilsson[1], Per Enblad[1], Elisabet Ohlin Sjöström[2], Anja Smits[4]

1 Department of Neuroscience, Neurosurgery, Uppsala University Hospital, Uppsala, Sweden, 2 Department of Immunology, Genetics and Pathology, Uppsala University Hospital, Uppsala, Sweden, 3 Department of Medical Sciences, Endocrine Oncology, Uppsala University Hospital, Uppsala, Sweden, 4 Department of Neuroscience, Neurology, Uppsala University Hospital, Uppsala, Sweden

* jimmy.sundblom@neuro.uu.se

## Abstract

### Background

Hemangioblastomas of the central nervous system are a prominent feature of von Hippel-Lindau-disease (vHL). Hemangioblastomas are known to secrete vascular endothelial growth factor (VEGF), suggesting a potential role of VEGF as a biomarker for tumor growth.

### Methods

Plasma VEGF samples from 24 patients with von Hippel-Lindau disease were analyzed by solid-phase proximity ligation assay (PLA). Levels were monitored over time together with numeric and volumetric CNS tumor burden, and compared to plasma VEGF levels in healthy controls.

### Results

The mean yearly progression in tumor volume was 65.5%. Yearly risk of developing one or several new CNS tumor(s) was 50%. No significant correlation between tumor burden and levels of VEGF was seen. VEGF levels in patients (31.55–92.04; mean 55.83, median 56.41) as measured by immunodetection in a solid-phase PLA did not differ significantly from controls (37.38–104.56; mean 58.89, median 54.12) (p = 0,266).

### Conclusion

The increase in total CNS tumor volume in vHL occurred in a saltatory manner. The risk of developing a new lesion was 50% per year. We found no evidence for VEGF secretion from CNS hemangioblastomas in vHL in circulating blood. Other potential biomarkers should be explored to assess progression of tumor burden in vHL.

**Data Availability Statement:** All relevant data are available from the OSF database (https://osf.io/2k5pn/).

**Funding:** The author(s) received no specific funding for this work.

**Competing interests:** The authors have declared that no competing interests exist.

## Introduction

Von Hippel-Lindau disease (vHL) is an autosomal dominant inherited disorder predisposing to development of vascular tumors of the central nervous system (CNS), retina, endocrine tumors, cystic lesions of the pancreas and kidneys, renal clear cell carcinoma (RCC) and several other tumors [1]. CNS hemangioblastoma is one of the most prominent features of vHL, with 90% of patients exhibiting this type of lesion [2]. They occur almost exclusively infratentorially, in the cerebellum, brain stem and spinal cord.

The most common cause of the disease is mutations of the *VHL* gene on chromosome 3, but the phenotype may exhibit in patients without mutations in the gene itself, probably due to mutations in regulatory sequences [3, 4]. The *VHL* gene acts as a tumor suppressor and its corresponding protein (pVHL) has two isoforms, a full length pVHL30 and pVHL19 generated by alternative translation. Both proteins are expressed in several types of tissue, regulating transcription of several growth factors, including angiogenic ones [5]. Clinical diagnosis of vHL is made when a patient presents with more than one vHL-associated lesion (including multiple CNS hemangioblastomas), or, in case of a positive family history, one single vHL-associated lesion. Genetic testing should be considered in patients even with a solitary vHL lesion [6]. Negative genetic testing in a patient fulfilling the diagnostic criteria does not exclude the patient from participating in a screening program. Nevertheless, the uptake in screening programs is dependent on compliance, which due to the cumbersome nature of the many investigations can be low [7].

Due to the range of different organ systems involved in vHL, and to the non-predictable nature of the disease, the optimal basis for care is surveillance by multidisciplinary teams. Application of a clinical screening program has been shown to decrease the risk of disabling or fatal tumor development in patients [8]. However, life expectancy in vHL is still low compared to other hereditary cancer syndromes, 52.5 years, with mortality mainly caused by CNS lesions and RCC [9]. Regarding CNS lesions, the current paradigm is surveillance of small tumors and microsurgical extirpation of cerebellar tumors when mass effect is evident, while brain stem and spinal cord tumors are surgically treated when symptoms occur (due to surgical morbidity associated with lesions in these regions) [10–12]. Of course, this strategy confers the risk that already developed symptoms will not regress after surgery. Also, small tumors may grow significantly between screening visits. Consequently, readily available biomarkers predicting growth of CNS hemangioblastomas would be a welcome addition to the surveillance program, to fine-tune individual management and possibly also to increase compliance to surveillance in patients. The aim of this study was to analyze circulating VEGF levels in relation to total growth rate and risk of developing new lesions.

## Materials and methods

### Participants

Twenty-three patients with genetically or clinically confirmed vHL (age 18–68; 14 female, 9 male; Table 1) were prospectively recruited from the Departments of Endocrine Oncology, Neurology or Neurosurgery at Uppsala University Hospital. Plasma samples were collected at yearly visits. VEGF was analyzed at least one timepoint in all 23 individuals.

Plasma samples from healthy control individuals (n = 29) were obtained at yearly visits. All samples were stored at -70 until analysis.

Most patients (n = 22) followed screening protocols stipulating yearly magnetic resonance imaging (MRI) studies, but seven patients were excluded from the radiological part of the study due to lack of adequate corresponding MRI studies.

**Table 1. Patients and clinical characteristics.**

| Patient | Mutation | Gender | Age at inclusion | Retinal hemangioma | Neuroendocrine tumor | Kidney lesion | Endolymphatic sac Tumor |
|---|---|---|---|---|---|---|---|
| VHL001 | nc194 C>T (p.Ser65Leu) | M | 36 | Y | N | Y | N |
| VHL002 | nc699 C>G (p.Cys162Trp) | F | 43 | Y | N | N | N |
| VHL003 | none | F | 32 | N | N | N | N |
| VHL004 | none | F | 65 | Y | N | N | N |
| VHL005 | ? | F | 41 | Y | N | N | N |
| VHL006 | nc699 C>G (p.Cys162Trp) | F | 22 | Y | N | N | N |
| VHL007 | nc699 C>G (p.Cys162Trp) | M | 19 | Y | N | Y | N |
| VHL008 | nc778 delG (frameshift) | M | 28 | Y | N | N | N |
| VHL009 | ? | F | 41 | Y | Y | Y | N |
| VHL010 | p26(?) | M | 21 | N | N | N | N |
| VHL011 | nc712 C>T (p.Arg167Trp) | F | 43 | Y | N | Y | N |
| VHL012 | nc712 C>T (p.Arg167Trp) | F | 44 | Y | Y | Y | N |
| VHL013 | none | F | 54 | N | N | N | N |
| VHL014 | deletion (complete) | F | 46 | Y | N | Y | N |
| VHL015 | none | F | 68 | N | Y | N | N |
| VHL016 | nc694 C>T (p.Arg161Stop | F | 23 | Y | N | N | N |
| VHL017 | nc694 C>T (p.Arg161Stop) | M | 19 | N | N | N | N |
| VHL018 | nc778 delG | M | 55 | Y | N | Y | N |
| VHL019 | nc233 A>G (p.Asn78Ser) | M | 45 | N | N | Y | N |
| VHL020 | nc713 G>A (p.Arg167Gln) | F | 38 | Y | N | N | N |
| VHL021 | nc712 C>T (p.Arg167Trp) | M | 43 | Y | Y | Y | N |
| VHL022 | nc505 delTACCC (frameshift) | F | 54 | Y | Y | Y | N |
| VHL023 | deletion (complete) | M | 18 | Y | N | N | N |

## Ethical considerations

The study was approved by the regional ethics review board. All participants provided written informed consent.

## Plasma VEGF analysis by solid-phase proximity ligation assay (PLA)

Affinity-purified polyclonal biotinylated antibodies against $VEGF_{165}$ were procured from R&D Systems (BAF293). Streptavidin-conjugated Biovic3 and Biovic5 were purchased from Avidomics. The forward primer (Biofwd), reverse primer (Biorev) and connector oligonucleotide were obtained from Integrated DNA Technologies (IDT). Biotinylated antibodies were immobilized on the Dynabeads MyOne T1 streptavidin coated beads (Thermo Fisher).

The probes were functionalized by conjugating single-stranded DNA molecules of approximately 60 nucleotides in length to antibodies via biotin-streptavidin conjugation.

For the preparation of PLA probes using streptavidin modified oligonucleotides, biotinylated antibodies, were separately mixed with 100 nM streptavidin-oligonucleotides (streptavidin-Biovic3 and streptavidin-Biovic5) at a 1:1 molar ratio. Solid-phase PLA was performed as described previously [13]. The assays commenced with the mixing of 45µl of each sample with the 5µl of the microparticle beads in microtiter wells and incubated for 1–1.5 h at RT under rotation. After the incubation, 50µl of PLA probe mixture at concentration of 500pM for each probe was added to each well. Subsequently, the microparticle beads were washed twice with washing buffer and 50µl of qPCR master mix (1x PCR buffer, 2.5mM MgCl2 (Invitrogen),

0.1μM concentration of each primer (Biofwd and Biorev) and connector oligonucleotide (Biosplint), 0.5X Sybr Green (Thermo Fischer Scientific), 0.08mM ATP, 0.2mM dNTPs (containing dUTP), 1.5 units of Platinum *Taq* polymerase, 0.02 units of T4 DNA ligase (30U/μl) (Sigma-Aldrich), 0.1 unit of uracil–DNA glycosylase (1U/μl)(Thermo Fischer Scientific)) were added to each well, and followed by detection of ligation product via qPCR performed on ABI 7900 (Thermo Fischer Scientific). For detailed description of the analysis, see S1 File.

## Data analysis

The recorded cycle treshold values for q-PCR data were further analyzed with Microsoft Excel. In addition, ImageJ software was used in the results analysis to determine limit of detection (LOD), lowest limit of quantification (LLOQ) and dynamic ranges for all assays. The LOD for the SP-PLAs was defined as the concentration of protein corresponding to $Ct_{LOD} = Ct_N − (2 \times S_N)$, where $Ct_N$ is the average Ct acquired for the background noise, and $S_N$ is the standard deviation of that value. The LLOQ for SP-PLA was defined as $Ct_N − (10 \times S_N)$.

## Radiological studies

Radiological studies were performed at different centres since the patients usually underwent MRI at regional hospitals before clinical visit at the vHL center. Thus, protocols differed slightly, but all cranial studies included T1 sequences with and without contrast and T2 TSE sequences. All spinal studies included sagittal T1 w/wo contrast, axial T1 w contrast for every significant tumor and sagittal T2 series (STIR/TSE).

Contrast-enhancing MRI lesions in the posterior fossa and spinal cord were assessed and counted in axial T1 series. Cystic portions associated with contrast enhancing lesions were assessed in axial T2 series. The slice thickness of the cranial MRI investigations was 1 mm for T1 series and 3 mm for T2 series. Slice thickness of spinal MRI investigations was 3 mm.

Very small lesions had to be detectable in at least two adjacent sections as well as confirmed in both coronal and sagittal series to be classified as a tumor.

Tumor volume was assessed by volumetry performed in a picture archiving and communication system (PACS) client (VueMotion Radiology Client; Carestream, Rochester NY). Central elongated T2 spine lesions deemed to represent syringomyelia was not included in cyst volumetry, while lateral/rounded spinal cystic lesions were included.

Each tumor was measured by two individuals separately to increase reliability. Very small tumors (diameter of <1mm, not measurable in more than two adjacent sections, making volumetric assessment impossible due to program limitations) were assigned a volume of 1 $mm^3$.

## Genetic analysis

All patient records were obtained and information regarding mutations in the vHL-gene was collected.

## Statistical analysis

VEGF levels between patients and controls were compared using students T-test. A p-value of <0.05 was considered significant.

The correlation between tumor volume and VEGF levels in patients with CNS hemangioblastomas was assessed using $R^2$-value, where a strong correlation is suggested by a value close to 1. Significance was further assessed by $chi^2$-testing, where a p-value <0.05 was considered significant.

**Table 2. VEGF levels at initial visit in patients with CNS lesions.**

| Patient no | No of CNS lesions | Total tumor volume(mm³) | Tumor volume including cysts (mm³) | VEGF (pM) |
|---|---|---|---|---|
| VHL002 | 13 | 675,37 | 675,37 | 38,50 |
| VHL005 | 4 | 129,00 | 129,00 | 42,85 |
| VHL006 | 15 | 275,81 | 275,81 | 41,80 |
| VHL007 | 4 | 934,17 | 1174,07 | 41,23 |
| VHL008 | 9 | 547,44 | 661,14 | 31,55 |
| VHL009 | 10 | 1140,77 | 1935,57 | 40,70 |
| VHL010 | 2 | 38,59 | 38,59 | 45,41 |
| VHL011 | 17 | 1202,20 | 2382,2 | 76,58 |
| VHL012 | 13 | 1527,23 | 1527,23 | 60,26 |
| VHL014 | 3 | 2169,84 | 2169,84 | 68,55 |
| VHL016 | 1 | 2,46 | 2,46 | 51,99 |
| VHL017 | 1 | 30,70 | 30,70 | 41,16 |
| VHL020 | 5 | 319,50 | 319,50 | 44,58 |
| VHL021 | 2 | 12,75 | 236,55 | 56,41 |
| VHL022 | 2 | 43,16 | 43,16 | 52,43 |
| VHL023 | 6 | 196,15 | 499,35 | 92,04 |

## Results

### Tumor burden and growth rate

The number of cranial hemangioblastomas at initial visit in the 16 patients examined by radiological investigations ranged from 0 to 17 (median 4), while the number of spinal hemangioblastomas ranged from 0 to 9. Total tumor volume ranged from 2,46 to 2169,84 mm² (Table 2).

Tumor volume and growth rate are visualized in Fig 1 for all patients undergoing more than one MRI. All patients who showed a decrease in tumor volume had undergone surgery to remove one or more tumors. No decrease in tumor volume was seen in patients not undergoing surgery. The yearly risk of developing a new tumor was 50%. The mean yearly increase in total tumor volume was 65.5%.

There was no difference in tumor progression between female and male participants.

### Mutations

Ten different known disease-causing mutations were found in the patients. The most common was Nc699C>G, causing a cysteine to tryptophane change at position 162, which was found in three patients. Two patients exhibited a complete deletion of the vHL gene (Table 1). Two patients had chosen not to undergo genetic testing.

Four patients diagnosed by clinical criteria did not exhibit any known disease-causing mutations in the VHL gene. None of these patients had, or developed, any CNS lesions.

### VEGF levels

Plasma VEGF levels did not differ between patients (31.55–92.04 pM; mean 55.83, median 56.41) and controls (37.38–104.56; mean 58.89, median 54.12) (p = 0,266) (Fig 2). Exclusion of patients without known mutations, or with no known CNS lesions, respectively, did not cause any significant changes in mean or median values.

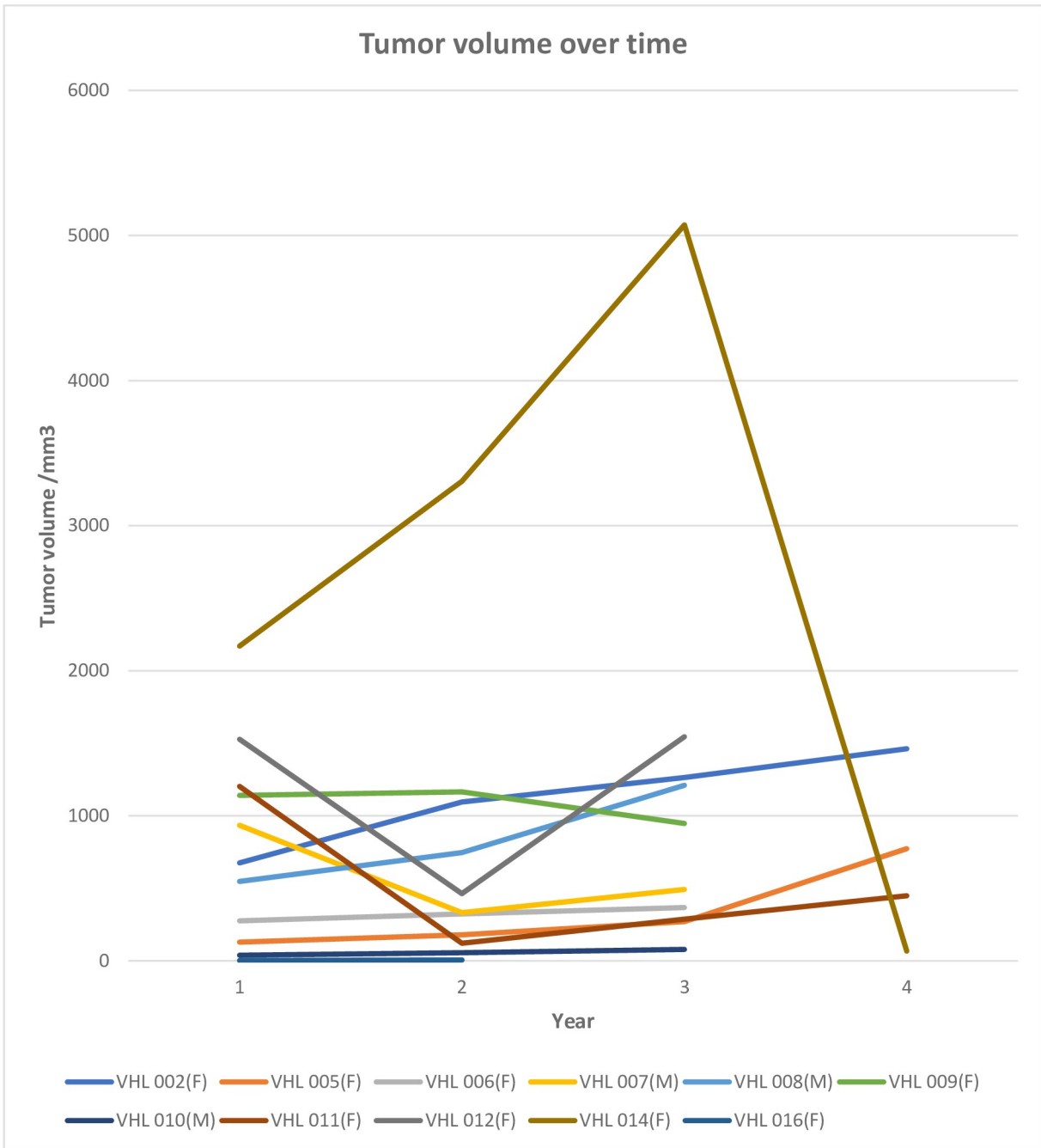

**Fig 1. Tumor volume increase over time.** Note the differing growth patterns. Dramatic increases seen most often pertained to increased size of one single tumor. Only patients with more than one completed MRI-investigation are shown. Decrease in volume was only seen in patients who had undergone surgery. VEGF was analyzed at initial visit (year 0) in all patients.

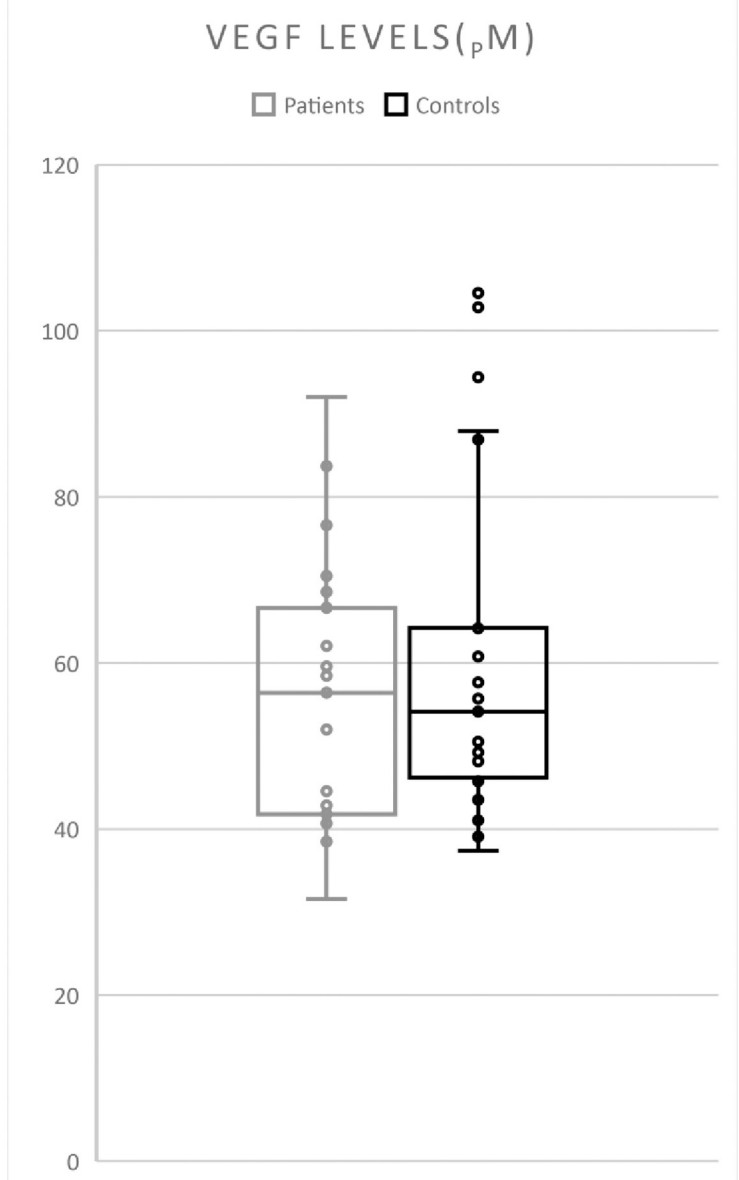

**Fig 2. VEGF levels in patients compared to controls.** No difference in VEGF levels was seen between groups.

There was no correlation found between levels of circulating VEGF and total tumor volume among patients harboring CNS lesions ($R^2$ = 0.0639) (Fig 3). Neither was there any correlation found when tumor associated cysts were included in the measurement ($R^2$ = 0,1221).

## Discussion

Biomarker development is of paramount importance in diseases, such as vHL, with a varying and unpredictable clinical course, in which surveillance of individual patients is the key for optimal clinical management. At present, the biomarkers used for vHL are standard blood sampling for endocrine and kidney function (including metanephrine for pheochromocytoma), radiological studies for solid tumors and cysts, and opthalmological examination for retinal lesions. Additional"wet" biomarkers that could function as surrogate markers for MRI,

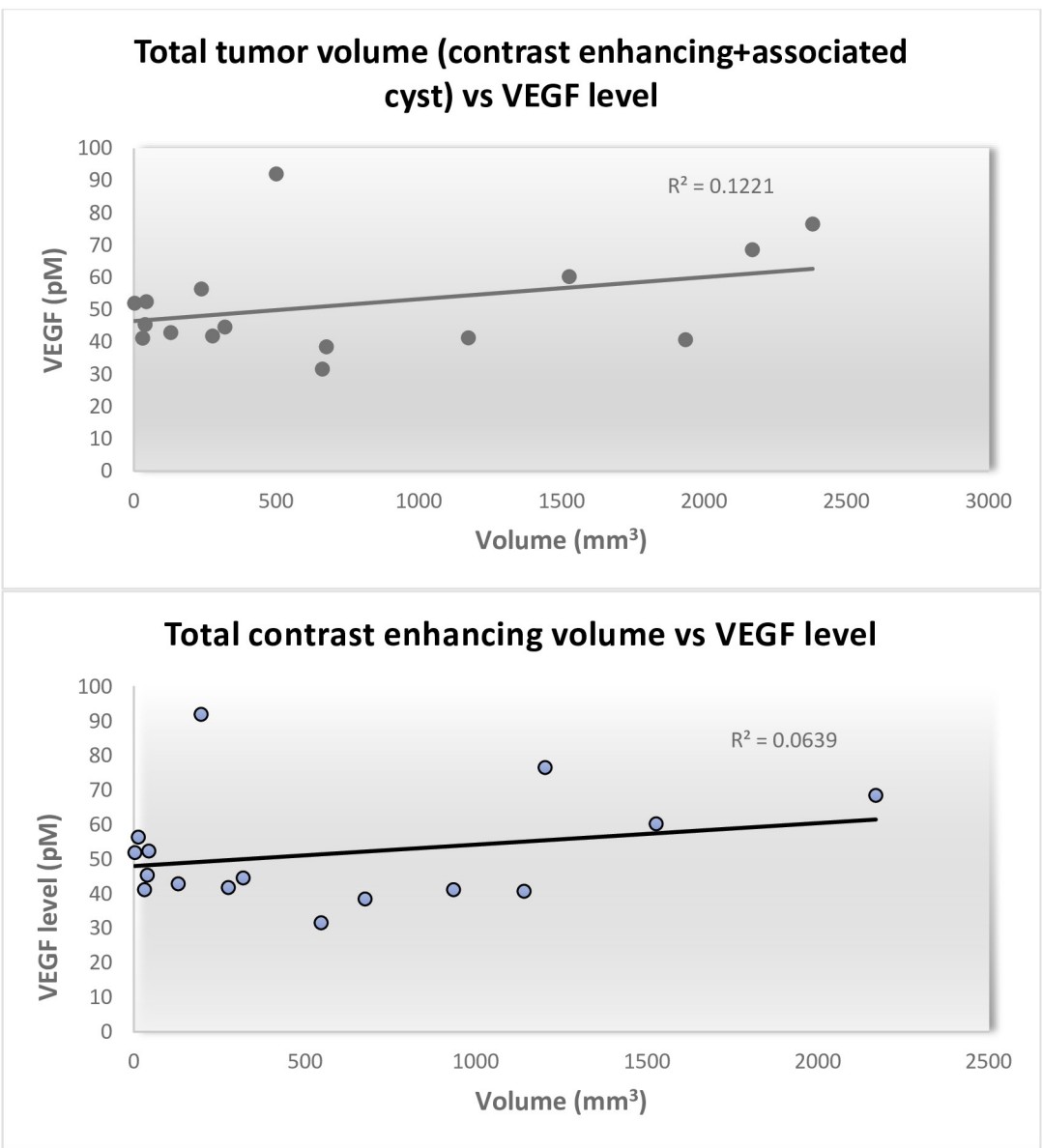

**Fig 3.** a. VEGF levels compared to total tumor volume including cystic portions at initial investigation. No significant correlation was seen. b. VEGF levels compared to only contrast-enhancing tumor volume. No significant correlation could be found. Excluding the one patient with largely elevated VEGF-levels as an outlier still did not suggest any significant correlation.

would be welcomed for vHL surveillance. The discovery of new biomarkers could also elucidate further pathways affected in the disease and suggest new avenues for treatment.

The case for VEGF as a biomarker in vHL is theoretically strong. pVHL creates complexes regulating ubiquitin-dependent proteolysis (E3 ubiquitin protein ligase), which among other interactions regulates hypoxia-inducible factor (HIF). HIF induces transcription of over 50 genes, including growth factors such as VEGF [14].

Hemangioblastomas are known to secrete VEGF in vitro [15]. There is grade 3 evidence for the use of angiogenesis inhibitors as salvage systemic therapy in vHL and/or sporadic hemangioblastomas, for instance interferon-beta [16], VEGF-targeting tyrosine kinase inhibitors

such as sunitinib [17] and semaxanib [18] and monoclonal antibodies such as bevacixumab [19]. Peripheral plasma VEGF levels were monitored and decreased in a study of senaxamib, but whether levels differed from control individuals was not investigated [18].

In the present study, we found no significant difference between vHL patients or control, and no evidence that VEGF levels in blood predict CNS lesion growth or novel CNS lesions. The present findings do not exclude that VEGF could have a predictive role as a biomarker in vHL. The half-time of VEGF in blood is notoriously short and it may well be that VEGF degrades very quickly in blood samples, making it unsuitable as a biomarker. Peripheral blood cells also secrete VEGF, levels of which can increase in many instances, including peripheral vascular disease, inflammation, and platelet activation [20, 21]. It is probable that the levels of circulating VEGF derived from peripheral blood cells are too high for any tumor-derived VEGF to significantly affect levels in peripheral blood.

Furthermore, the nature of this study prevents samples of the same age from being analyzed and thus some samples have been stored longer which can affect protein stability. Repeated freeze thawing can also influence VEGF sample concentration [22] and although the difference in the number of freeze-thaw cycles for these samples is small, it may nevertheless have some impact.

VEGF secreted in central tumors would predominantly circulate in the brain interstitial fluid and cerebrospinal fluid (CSF), before being passed on to the blood compartment. Analyzing levels of VEGF in CSF could possibly resolve this issue, and would probably be a superior method, but due to invasiveness and the predisposition of hemangioblastomas for the posterior fossa, lumbar puncture and CSF sampling is not a viable strategy for the disease, due to the inherent risks of lumbar puncture in patients with cerebellar and/or brainstem lesions [23]. VEGF secreted from tumors also differs in isoforms from VEGF from other tissues, and it is possible that investigating different isoforms may yield other results, since the levels of tumor-secreted VEGF might be insignificant in comparison with that secreted for instance by muscle tissue [24]. Neither did VEGF levels in individual patients increase in a predictable manner with increased tumor burden, another reason to suggest that the use of VEGF as a biomarker in clinical studies is probably not recommended.

Tumor cysts, cell sparse and reactive, contain fluid similar in chemical composition to blood and not tumor tissue [25]. The presence or absence of cystic portions of the tumors was not reflected in plasma VEGF levels. Hemangioblastoma cysts are associated with faster growth and more symptoms, and younger patients seem more prone to developing fast-growing cysts [26].

Another limitation of this study is the fact that retinal angiomas were not considered, but in comparison to CNS hemangioblastomas the volume of these must be considered very small.

The increase in total CNS tumor volume over time in VHL patients seems to progress in a saltatory manner, much as individual tumors [2, 26]. Growth rate increases with increased tumor volume, a feature consistent with growth characteristics of other types of tumors, thus assuming exponential growth [27]. This can be explained with the increased proliferation rate of a larger tumor cell population. The risk of developing new CNS tumors in this material was high compared to earlier studies [2, 26], but it should be noted that the follow-up time is quite short. Furthermore, it is conceivable that the patients with low-active disease did not participate fully in the screening program and thus did not enter the study or were excluded from this part of the study due to lack of follow-up investigations. Nevertheless, this data adds valuable information which can improve counseling and support for clinical decision-making for vHL patients.

## Conclusion

VEGF as a plasma biomarker in vHL does not show promise as a plasma biomarker for detecting disease progression in this prospective study. Other potential biomarkers should be explored, by means of exploratory approaches such as proteomics and metabolomics.

The total CNS tumor burden increases with increased total tumor volume and exhibit saltatory growth much as individual tumors. The risk of developing a new CNS lesion in one year was found to be 50% in this material.

## Supporting information

**S1 File. A detailed technical description of the PLA analysis is provided in the supporting information.**
(DOCX)

## Acknowledgments

SciLifeLab; PLA and single cell proteomics unit, Uppsala, Sweden was of important assistance in performing the PLA analysis.

## Author Contributions

**Conceptualization:** Jimmy Sundblom, Pelle Nilsson, Per Enblad, Elisabet Ohlin Sjöström, Anja Smits.

**Data curation:** Madelene Braun.

**Formal analysis:** Jimmy Sundblom, Tor Persson Skare, Olivia Holm, Elisabet Ohlin Sjöström.

**Funding acquisition:** Per Enblad, Anja Smits.

**Investigation:** Jimmy Sundblom, Tor Persson Skare, Olivia Holm, Madelene Braun, Elisabet Ohlin Sjöström, Anja Smits.

**Methodology:** Staffan Welin, Pelle Nilsson, Per Enblad.

**Project administration:** Staffan Welin, Anja Smits.

**Resources:** Staffan Welin, Pelle Nilsson, Per Enblad, Anja Smits.

**Supervision:** Jimmy Sundblom, Staffan Welin, Pelle Nilsson, Per Enblad, Elisabet Ohlin Sjöström, Anja Smits.

**Visualization:** Jimmy Sundblom, Tor Persson Skare.

**Writing – original draft:** Jimmy Sundblom, Olivia Holm.

**Writing – review & editing:** Staffan Welin, Madelene Braun, Pelle Nilsson, Per Enblad, Anja Smits.

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
