## [Decision Letter · Decision Letter 0]

22 Jul 2022

PONE-D-22-00869Central nervous system hemangioblastomas in von Hippel-Lindau disease: total growth rate and risk of developing new lesions not associated with circulating VEGF levelsPLOS ONE

Dear Dr. Sundblom,

Thank you for submitting your manuscript to PLOS ONE. After careful consideration, we feel that it has merit but does not fully meet PLOS ONE’s publication criteria as it currently stands. Therefore, we invite you to submit a revised version of the manuscript that addresses the points raised during the review process.

We look forward to receiving your revised manuscript.

Kind regards,

Vanessa Carels

Staff Editor

PLOS ONE

Journal Requirements:

Reviewers' comments:

Reviewer's Responses to Questions

**Comments to the Author**

1. Is the manuscript technically sound, and do the data support the conclusions?

Reviewer #1: Yes

Reviewer #2: Partly

Reviewer #3: No

2. Has the statistical analysis been performed appropriately and rigorously? 

Reviewer #1: Yes

Reviewer #2: Yes

Reviewer #3: Yes

3. Have the authors made all data underlying the findings in their manuscript fully available?

Reviewer #1: Yes

Reviewer #2: No

Reviewer #3: Yes

4. Is the manuscript presented in an intelligible fashion and written in standard English?

Reviewer #1: Yes

Reviewer #2: Yes

Reviewer #3: Yes

5. Review Comments to the Author

Reviewer #1: Hemangioblastomas are known to secrete vascular endothelial growth factor (VEGF), suggesting a potential role of VEGF as a biomarker for tumor growth. The authors looked plasma VEGF samples from 24 patients with von Hippel-Lindau disease were

analyzed by solid-phase proximity ligation assay (PLA). Levels were monitored over time together with numeric and volumetric CNS tumor burden, and compared to plasma VEGF levels in healthy controls. They found no difference not surprisingly. The VEGF level may be different locally or in CSF and major changes are unlikely systematically this negative study verifies this suggestion. Hopefully they can look at CSF level in the future and over various time periods although this may be difficult to have patients consent to. I would like the authors to clarify the tumor volume change in patient #4 in Figure #1. Add the volumes with SD over time year 1,2,3, and 4 and also indicate the volumes for female and male with st deviation as well.

Reviewer #2: The authors have presented evidence for the lack of association between circulating VEGF levels and CNS tumor burden in VHL patients. The authors developed a solid phase PLA to measure VEGF165a. They then measured the serum VEGF levels in VHL patients (clinical criteria) and normal volunteers. They report no difference in serum VEGF levels with VHL disease or with CNS tumor burden.

As with other publications that have reported on paracrine/autocrine actions for pro-angiogenic factors in VHL (PMID: 27748427), the authors report no detectable difference with VHL disease or with increasing tumor burden. The manuscript is well written and covers the prior literature well. However, I have some comments for the authors:

1. The authors do not discuss the rationale for developing a brand new assay to measure circulating VEGF165a levels. Can they explain why off-the-shelf ELISA assays were not used? How was the solid phase PLA validated? What were the positive and negative controls used?

2. Another major issue is the single time-point measurement for VEGF in this study. The study design will inherently fail to detect changes with time/intervention. There is at least some evidence from animal studies that while VEGF levels are not different with/without VHL-/- tumors, interventions can lead to decreased VEGF levels (PMID: 30497198). This finding is like the clinical data (senaxamib study) cited by the authors. Can the authors explain the rationale for a single time point measurement?

3. Did the authors use ellipsoid volume or volumetric segmentation to arrive at tumor volumes?

4. The authors need to correct the text when citing the range of tumor numbers and tumor volumes in Results. Table 2 suggests that the lowest tumor volume is 2,46 mm3 while in the main text, the authors cite 0 mm3.

5. The authors make incorrect statements regarding the lack of information about increase in tumor growth in VHL in Discussion. Information about saltatory growth and the unpredictability has been well described in the prior literature with much larger cohorts.

6. In Figure 1, the authors need to mark the time-point for each patient when the serum VEGF sampling was done.

7. In Figure 2, individual values need to be shown (rather than a summary box-whisker plot).

Reviewer #3: Sundblom et al aimed to study the relations between circulating VEGF levels and HB development/progression in patients with VHL.

The idea is generally sound, but highly simplistic, as the increase in VEGF levels is cellular, and depends on the full loss of VHL in the tumors. Hence, the chances to find such an association is very low, to begin with. Moreover, the sample size is extremely small (although dealing with rare diseases, there are much larger cohorts and multi-center collaboration would be expected on such topics).

The paper could have been written better, both in terms of clinical accuracy and structure. The methods are too detailed in regard to the lab techniques, while the MRI interpretation is quite poor. This is especially important in VHL, where the T2 reflects the cystic portion of the tumors and is not described at all. While this may be explained by the aim to correlate with VEGF, the cystic part, together with the edema, might be at least as important as the solid part. Thus the clinical utility of VEGF must be tested vs. these components as well.

The diagnostic criteria for VHL include multiple hemangioblastomas (this might be the 1st one, in fact), and it is surprising that this criterion of all is not mentioned in the introduction...

The genetic data should be written more clearly, with both cDNA variant detail and protein-levels changes (VHL c.499G>C, p.R167W, for example).

The results should be thoroughly revised, written in a more fluent order, with results supporting the text, and not in isolated short phrases (for example line 232).

6. PLOS authors have the option to publish the peer review history of their article (what does this mean?). If published, this will include your full peer review and any attached files.

Reviewer #1: No

Reviewer #2: No

Reviewer #3: No

---

## [Author Response · Author response to Decision Letter 0]

13 Sep 2022

Regarding reviewer #1

As commented in the manuscript, we also would like to assess VEGF levels in CSF of vHL patients in the future. The nature of the lesions may make this somewhat difficult, but we hope to be able to address this in the future.

The tumor volume change in fig 1 regarding patient 14 (we presume that the 4 mentioned applies to pt 14, since pat 4 is not included in this analysis due to lack of follow-up investigations) is due to surgical intervention. This is hopefully made clearer in the figure legend in the revised manuscript. Since the total tumor volume is an absolute value, no SD is applicable. Gender has been added to the pt numbers in the figure.

Regarding reviewer #2

1. The PLA method was set up by our lab to possibly increase throughput. It was previously validated using an ELISA kit, and select samples in this study were analyzed with an ELISA with similar results. 

2. This study was designed to address whether the total CNS tumor volume is reflected in circulating VEGF. We will hopefully be able to continue working and address this very relevant question in the future, but a clinically useful biomarker should reflect the tumor burden at a single time point. 

3. Volumetric segmentation was used. This has been made clearer in the methods section.

4. This mistake has been corrected-

5. The statement has been amended. References to relevant articles (already cited) has been added.

6. The VEGF sampling was done at initial visit (year 1). This has been added to the figure legend.

7. The figure has been adjusted accordingly.

Regarding reviewer #3

The critique regarding the small number of patients is relevant, and we wish to start collaborating with other centers to increase the sample size in the future. 

In the revised manuscript, the details regarding the laboratory methods have been somewhat shortened. We have also rewritten and expanded on the radiology portion of the methods section. We have also performed volumetry of associated tumor cysts in patients harboring these lesions and investigated whether total tumor volume including cystic parts correlates with VEGF levels. A new figure has been added and the results are presented in the text. Since no correlation was seen this does not change the message of the manuscript. Although clinically important, volumetric measurement of edema is not technically reliable enough to include.

The diagnostic criteria of multiple hemangioblastomas for vHL was mentioned in the introduction (more than one vHL-associated lesion), but we have highlighted this instance further.

The genetic data has been updated as suggested.

The results section has been revised and will now hopefully read in a more fluent way.

---

## [Decision Letter · Decision Letter 1]

6 Nov 2022

PONE-D-22-00869R1Central nervous system hemangioblastomas in von Hippel-Lindau disease: total growth rate and risk of developing new lesions not associated with circulating VEGF levelsPLOS ONE

Dear Dr. Sundblom,

Thank you for submitting your manuscript to PLOS ONE. After careful consideration, we feel that it has merit but does not fully meet PLOS ONE’s publication criteria as it currently stands. Therefore, we invite you to submit a revised version of the manuscript that addresses the points raised during the review process. Please submit your revised manuscript by Dec 21 2022 11:59PM. If you will need more time than this to complete your revisions, please reply to this message or contact the journal office at plosone@plos.org. Please include the following items when submitting your revised manuscript:A rebuttal letter that responds to each point raised by the academic editor and reviewer(s). You should upload this letter as a separate file labeled 'Response to Reviewers'.A marked-up copy of your manuscript that highlights changes made to the original version. You should upload this as a separate file labeled 'Revised Manuscript with Track Changes'.An unmarked version of your revised paper without tracked changes. You should upload this as a separate file labeled 'Manuscript'.If applicable, we recommend that you deposit your laboratory protocols in protocols.io to enhance the reproducibility of your results. Protocols.io assigns your protocol its own identifier (DOI) so that it can be cited independently in the future. For instructions see: https://journals.plos.org/plosone/s/submission-guidelines#loc-laboratory-protocols. Additionally, PLOS ONE offers an option for publishing peer-reviewed Lab Protocol articles, which describe protocols hosted on protocols.io. Read more information on sharing protocols at https://plos.org/protocols?utm_medium=editorial-email&utm_source=authorletters&utm_campaign=protocols.

We look forward to receiving your revised manuscript.

Kind regards,

Prashant Chittiboina

Guest Editor

PLOS ONE

Journal Requirements:

Reviewers' comments:

Reviewer's Responses to Questions

**Comments to the Author**

1. If the authors have adequately addressed your comments raised in a previous round of review and you feel that this manuscript is now acceptable for publication, you may indicate that here to bypass the “Comments to the Author” section, enter your conflict of interest statement in the “Confidential to Editor” section, and submit your "Accept" recommendation.

Reviewer #1: All comments have been addressed

Reviewer #3: (No Response)

2. Is the manuscript technically sound, and do the data support the conclusions?

Reviewer #1: Yes

Reviewer #3: Yes

3. Has the statistical analysis been performed appropriately and rigorously? 

Reviewer #1: Yes

Reviewer #3: Yes

4. Have the authors made all data underlying the findings in their manuscript fully available?

Reviewer #1: Yes

Reviewer #3: Yes

5. Is the manuscript presented in an intelligible fashion and written in standard English?

Reviewer #1: Yes

Reviewer #3: Yes

6. Review Comments to the Author

Reviewer #1: They have addressed my concerns. They add the requested information and the manuscript should be considered for publication.

Reviewer #3: The manuscript writing improved significantly, although a few minor issues remained.

My main criticism for the study is yet in its core hypothesis – most of the VEGF in the circulation derive from peripheral blood cells, and the chances to see hemangioblastoma-derived VEGF in a meaningful concentration is extremely low. To prove that, even comparing patients with various VHL-related, VEGF-dependent tumors, to controls, shown no difference.

In my view, the authors might want to discuss this point in the discussion.

In addition, please revise the variants descriptions in Table 1 – the 167 codons is described as nc499 in VHL011 and nc712 in VHL012.

Minor

Editing, typos – therefor -> therefore, missing full stop in the abstract, changing fonts in the abstract.

Genetic analysis – VHL gene should be written in capital letters and in Italics.

7. PLOS authors have the option to publish the peer review history of their article (what does this mean?). If published, this will include your full peer review and any attached files.

Reviewer #1: No

Reviewer #3: No

---

## [Author Response · Author response to Decision Letter 1]

9 Nov 2022

Reviewer #3: The manuscript writing improved significantly, although a few minor issues remained.

My main criticism for the study is yet in its core hypothesis – most of the VEGF in the circulation derive from peripheral blood cells, and the chances to see hemangioblastoma-derived VEGF in a meaningful concentration is extremely low. To prove that, even comparing patients with various VHL-related, VEGF-dependent tumors, to controls, shown no difference.

In my view, the authors might want to discuss this point in the discussion.

-The discussion has been updated and references added to stress this important issue.

In addition, please revise the variants descriptions in Table 1 – the 167 codons is described as nc499 in VHL011 and nc712 in VHL012.

-This oversight has been corrected

Minor

Editing, typos – therefor -> therefore, missing full stop in the abstract, changing fonts in the abstract.

Genetic analysis – VHL gene should be written in capital letters and in Italics.

-These typos and others has been amended.

---

## [Editor Report · Decision Letter 2]

11 Nov 2022

Central nervous system hemangioblastomas in von Hippel-Lindau disease: total growth rate and risk of developing new lesions not associated with circulating VEGF levels

PONE-D-22-00869R2

Dear Dr. Sundblom,

We’re pleased to inform you that your manuscript has been judged scientifically suitable for publication and will be formally accepted for publication once it meets all outstanding technical requirements.

Kind regards,

Prashant Chittiboina

Guest Editor

PLOS ONE

---

## [Editor Report · Acceptance letter]

17 Nov 2022

PONE-D-22-00869R2 

Central nervous system hemangioblastomas in von Hippel-Lindau disease: total growth rate and risk of developing new lesions not associated with circulating VEGF levels 

Dear Dr. Sundblom:

I'm pleased to inform you that your manuscript has been deemed suitable for publication in PLOS ONE. Congratulations! Your manuscript is now with our production department. 

Kind regards, 

on behalf of

Dr. Prashant Chittiboina 

Guest Editor

PLOS ONE